# Reliability of Prediction Models for the Functional Classification of a Sinusoidal Intraocular Lens Depending on Pupil Diameter

**DOI:** 10.3390/diagnostics15192446

**Published:** 2025-09-25

**Authors:** Diego Montagud-Martínez, Walter D. Furlan, Vicente Ferrando, Manuel Rodríguez-Vallejo, Joaquín Fernández

**Affiliations:** 1Centro de Tecnologías Físicas, Universitat Politècnica de València, 46022 Valencia, Spain; viferma1@upv.es; 2Departamento de Óptica y Optometría y CC de la Visión, Universitat de València, 46022 Valencia, Spain; walter.furlan@uv.es; 3Qvision, Department of Ophthalmology of VITHAS Almería Hospital, 04120 Almería, Spain; manuelrodriguezid@qvision.es (M.R.-V.); joaquinfernandezoft@qvision.es (J.F.)

**Keywords:** intraocular lens, visual acuity, prediction model, modulation transfer function, functional classification, optical bench

## Abstract

**Background:** To assess the agreement among prediction models for the functional classification of intraocular lenses (IOLs) and discuss their limitations in evaluating pupil dependency of a sinusoidal IOL. **Methods:** An ISO-compliant optical bench setup with modifications to characterize the modulation transfer function area (MTFa) across pupil diameters from 1.5 to 5.5 mm was used to measure the Acriva Trinova Pro C Pupil Adaptive IOL. Six prediction models (Vega et al., 2018, Fernández et al., 2019, Alarcón et al., 2016, Armengol et al., 2020 were applied to estimate visual acuity defocus curves from MTFa and functional classification based on the depth-of-field (DOFi) and the increase in visual acuity (ΔVA) from intermediate to near. **Results:** Defocus curves for all prediction models consistently demonstrated a Full-DOFi response (>2.3 D at 0.2 logMAR), with differences in ΔVA emerging across pupil diameters. Continuous decreases (ΔVA < 0.05 logMAR) were observed at pupil diameters <2.5 mm, while Smooth transitions (ΔVA from 0.05 to 0.14 logMAR) occurred between 2.5–3.0 mm for all models except for Vega. At pupil diameters >3.5 mm, most models transitioned to a Steep classification (ΔVA ≥ 0.14 logMAR), except Fernández, which remained Smooth, and Armengol 2020a, which shifted to Steep at 4.0 mm. **Conclusions:** Visual acuity prediction models provide useful means of reporting optical bench data in clinically familiar metrics. However, outcomes should be interpreted with caution as functional classifications can vary depending on the optical bench setup and prediction model used.

## 1. Introduction

Intraocular lenses (IOLs) are implanted to replace the crystalline lens following cataract surgery, restoring visual clarity and improving quality of life by regaining transparency of the eye’s optical media [1]. Traditional monofocal IOLs effectively restore distance vision when targeted for emmetropia; however, patients usually remain dependent on spectacles for intermediate and near visual tasks [2]. To overcome this limitation, simultaneous vision IOLs (SVLs) have emerged, providing extended visual acuity ranges to facilitate spectacle independence at multiple distances [3].

However, classifying the performance of these IOLs requires careful consideration of the metrics employed. Optical laboratory studies commonly use the term depth-of-focus, defined as the permissible variation in image distance of an optical system without significant loss in sharpness [4]. In clinical practice, however, visual performance is assessed through visual acuity defocus curves (VADC), involving lenses of varying power to simulate viewing at multiple distances [4,5]. Given the differing contexts and methodologies, scientific societies highlighted the need for clear terminological distinction: depth-of-focus should be reserved for laboratory (optical bench) measurements, whereas depth-of-field (DOFi) should specifically denote clinically measured visual acuity outcomes [6,7].

Clinically, DOFi derived from defocus curves quantifies the functional vision range corresponding to a specific visual acuity threshold, commonly 0.2 or 0.3 logMAR. Based on DOFi values and defocus curve shape, IOLs are classified functionally into distinct categories. For example, Partial-DOFi lenses display moderate functional ranges (<2.3 D), further categorized as Narrowed, Enhanced, or Extended. In contrast, Full-DOFi lenses demonstrate broader functional ranges (≥2.3 D) and are subdivided by the steepness of their acuity profiles into Continuous, Smooth, or Steep types [7,8].

Nevertheless, clinical assessments are frequently constrained by variability factors, notably pupil diameter, complicating precise functional categorization [9]. When direct clinical measurements are unavailable or limited, laboratory evaluations, such as optical bench testing, offer valuable supplementary data. Metrics such as the Modulation Transfer Function area (MTFa), known to correlate well with clinical visual acuity, provide critical insights into anticipated visual performance [10,11,12,13]. However, these optical metrics remain challenging for clinicians to interpret directly. Therefore, computational prediction models as recommended by the ANSI Z80.35 and ISO 11979−2 standards [14,15] have become crucial in linking laboratory findings with clinical practice by translating complex laboratory data into clinically relevant outcomes [16].

However, these prediction models are not without limitations, which can introduce biases between clinical outcomes and those predicted in the laboratory. The purpose of this study is to evaluate the agreement among existing prediction models that estimate clinical VADC based on optical bench metrics [10,11,12,13], specifically assessing their reliability and consistency in evaluating pupil dependency, defined as the change in functional classification resulting from variations in pupil diameter, of a multifocal IOL with a sinusoidal diffractive profile. In addition to this primary aim, we discuss limitations that may introduce biases between predicted and observed clinical outcomes reported in the literature. Given that the industry often supports IOL performance claims with these predicted defocus curves, this study provides clearer guidance on the limitations and key considerations for interpreting laboratory-based VADC.

## 2. Materials and Methods

### 2.1. Intraocular Lens

The Acriva Trinova Pro C Pupil Adaptive IOL (VSY Biotechnology, Maltepe İstanbul, Turkey) is a single-piece, foldable lens designed to optimize vision across multiple focal distances. The lens features an asymmetric biconvex optic structure with a diffractive anterior surface based on Sinusoidal Vision Technology^®^ and an aspheric posterior surface inducing a spherical aberration (SA) of −0.1 μm at 6 mm. The diffractive zone spans from 1.40 mm to 6.0 mm, providing light distribution for far (order −1), intermediate (order 0), and near (order +1) vision. The manufacturer describes employing ‘Pupil Adaptive’ technology, which modulates light distribution dynamically based on pupil size. At a 3 mm aperture, it delivers 43% of the available light to far focus, 21% to the intermediate focus, and 36% to the near focus. As the aperture increases, the lens favors the intermediate focus over the near focus, enhancing functional vision in varying lighting conditions [17]. The addition powers are labeled as +1.80 D for intermediate and +3.60 D for near vision at the IOL plane. The material composition is hydrophilic acrylic with 25% water content, a refractive index of 1.46, and an Abbe number of 58 [18].

### 2.2. Measurement Procedure

Our experimental device is based on a custom-made image-forming system that complies with the requirements of the International Organization ISO 11979−2, 2014 standards [19]. The system, fully controlled by LabView 2021 Professional^®^ software (National Instruments, Austin, TX, USA), allows for obtaining various image quality metrics of an MIOL for different defocus values, pupil diameters, and wavelengths through an automated procedure [20].

A representation of the optical setup is shown in Figure 1. The illumination system provides a collimated beam from a white LED (Thorlabs MCWHL5, Thorlabs Inc., Newton, NJ, USA). In this experiment, this beam was sequentially filtered with five different chromatic filters, each with a 10 nm bandwidth and centered at 450 nm, 500 nm, 550 nm, 600 nm, and 650 nm, respectively (FB450−10 to FB650−10, Thorlabs Inc., Newton, NJ, USA).

Through-focus MTFa curves were obtained using a 30 μm pinhole as the test object. The pinhole was mounted on a motorized translation stage (Thorlabs LTS300/M, travel range 300 mm, accuracy 5 μm). Consequently, different vergences in the range of +1.0 D to −4.0 D were achieved using a Badal lens (focal length f′ = 160 mm). An adjustable iris diaphragm (ESKMA Optics; 995−1912−N) was placed at the image focal plane of this lens. This plane was further projected onto the anterior surface of the IOL under evaluation using a 4f system composed of lenses L1 and L2 (f1 = f2 = 100 mm). In this way, the variable iris acts as the pupil of the artificial eye.

The model eye consisted of an artificial cornea, the ISO 1 cornea [21], an achromatic lens with corrected SA (Melles Griot LA034, 27.8 D, Thorlabs Inc., Newton, NJ, USA), and a wet cell filled with saline solution, where the IOL under test was placed.

To obtain the images formed by the artificial eye, a CMOS camera (EO 5012C; Edmund Optics, Barrington, NJ, USA) attached to a microscope with was used as an artificial retina.

Using the device illustrated in Figure 1, the calculation of the MTFa curves was performed through the following steps. First, the PSFs were obtained for the five specified wavelengths described above. These PSFs were Fourier transformed to derive the corresponding MTFs, which were then integrated over the frequency range of 0 to 50 lp/mm to calculate the MTFa values. The MTFa values were subsequently weighted using the V(λ) function coefficients: 0.038 for 450 nm, 0.323 for 500 nm, 0.995 for 550 nm, 0.631 for 600 nm, and 0.107 for 650 nm, to obtain the polychromatic MTFa. The MTFa provided numerical values, which were then represented at intervals of 0.1 D over a range from +1.0 D to −4.0 D.

### 2.3. Prediction Models

Six prediction models were employed to estimate visual acuity along the DOFi based on the MTFa, calculated by integrating the MTF curve from 0 to 50 lp/mm, and scaling the defocus step range to simulate the defocus curve measured with defocus lenses at the spectacle instead of the IOL plane. The equations for these models are provided in Appendix A, and their key characteristics are summarized below:Vega et al. (2018) evaluated IOLs with a power of 20 D using an optical bench test under monochromatic green light. The MTF was measured using a four-slit pattern. A corneal model induced SA of +0.27 µm for a 6 mm of pupil diameter, while measurements were conducted with a 3 mm of pupil diameter at the IOL plane. The model, based on an exponential decay function for predicting monocular visual acuity, included both Full and Partial-DOFi IOLs [10].Fernández et al. (2019) conducted ray-tracing simulations under monochromatic green light using a single Full-DOFi IOL (21.5 D). A corneal model induced SA of +0.24 µm for a 6 mm optical surface. Simulations were performed for pupil diameters ranging from 2.5 mm to 4 mm in 0.5 mm increments. Their predictive models used both linear (2019a) and exponential decay (2019b) functions to estimate monocular visual acuity and contrast sensitivity defocus curves [11].Alarcon et al. (2016) obtained optical bench measurements of the MTF using the line spread function under polychromatic light for both Full and Partial-DOFi IOLs (20 D). An average corneal model was used to reproduce typical SA with a pupil diameter of 3 mm. The predictive model applied the MTFa raised to the power of −1. The model by Alarcon et al. differed from the others in predicting binocular rather than monocular visual acuity [12,22].Armengol et al. (2020) provided two predictive models under polychromatic light for estimating monocular visual acuity: the first utilized an inverse proportional function (Armengol 2020a), and the second applied an exponential decay function (Armengol 2020b). These models were developed using measurements from two Full and one Partial-DOFi IOLs (20 D) on an optical bench on which the MTF was measured using a four-slit pattern. A corneal model induced ±0.17 µm of SA for a 5 mm of pupil diameter, with measurements obtained for PDs of 3 mm and 4.5 mm [13].

The six prediction models differ in light sources, pupil diameters, and types of lenses analyzed. Vega and Fernández used monochromatic green light [10,11], while Alarcon and Armengol used polychromatic light [12,13]. Pupil diameters varied, with Vega using 3 mm [10], Fernández simulating 2.5–4 mm [11], Alarcon fixing 3 mm [12], and Armengol measuring 3 mm and 4.5 mm [13]. In addition, all prediction models included both Full and Partial-DOFi IOLs, except for the Fernández model, which focused exclusively on a single Full-DOFi IOL [10,11,12,13]. The shapes corresponding to these functional classifications are illustrated in Figure 2a, while Figure 2b presents the relationship between visual acuity and MTFa for each specific prediction models described above and detailed in Appendix A. Figure 2a demonstrates how the various prediction equations yield distinct visual acuity outcomes depending on different MTFa values.

### 2.4. Data Analysis

Outcomes were presented as VADC for pupil diameters ranging from 1.5 to 5.5 mm in 0.5 mm increments for each prediction model. Functional classification was used to describe the estimated VADC, distinguishing between two main responses based on the DOFi. The DOFi is defined as the dioptric range from 0 D to the defocus cut-off at 0.2 or 0.3 logMAR [6,7]. IOLs with an RoF ≥ 2.3 D at 0.2 logMAR are classified as Full-DOFi, whereas those with an DOFi < 2.3 D are considered Partial-DOFi. Among Full-DOFi IOLs, three subtypes are identified based on the visual acuity slop from intermediate to near distances: ‘Continuous’ decrease for differences <0.05 logMAR, ‘Smooth’ improvement for differences between 0.05 and 0.14 logMAR, and ‘Steep’ improvement for differences ≥0.14 logMAR [6,7].

## 3. Results

Figure 3 shows the defocus curves estimated for the different models and pupil diameters. In all cases, the VADC exhibited a Full-DOFi response, exceeding 2.3 D at 0.2 logMAR. However, the primary differences were observed in the variations of visual acuity from intermediate to near distances (Table 1). A Continuous decrease in visual acuity (<0.05 logMAR) was observed for all models with pupil diameters below 2.0 mm, and even at 2.5 mm for the Vega model. For pupil diameters between 2.5 and 3.0 mm, all models except Vega demonstrated a Smooth transition of visual acuity (0.05–0.14 logMAR). The Fernández models consistently showed lower magnitudes of differences between intermediate and near distances compared to other models, particularly at a pupil diameter of 3.0 mm, where the difference was 0.07 logMAR versus >0.11 logMAR for the other models. Beyond 3.5 mm, the magnitude of differences generally increased for all models except for Fernández model, resulting in a classification of Steep for the defocus curve in the remaining models (also except Armengol 2020a starting the Steep classification at 4 mm). This observed increase in differences and the corresponding shifts in classification suggest a pupil-dependent IOL response, with the functional classification altering from Smooth to Steep for pupil diameters above 3.5 mm in all models except for the Fernández model.

## 4. Discussion

Scientific publications on the optical characterization of new IOLs have significantly contributed to their increasing popularity over the past decade. Initially, these characterizations relied on experimental optical bench setups conducted by researchers [23,24]. More recently, the development and use of commercial devices for optical characterization have facilitated this process [25,26]. These advancements have greatly enhanced the understanding of new IOLs prior to the availability of clinical studies.

The metrics commonly reported in such publications are often challenging for clinicians to interpret, as they are more familiar with psychophysical metrics like visual acuity and contrast sensitivity, which are commonly used in daily clinical practice. To address this gap, prediction models have been developed to estimate psychophysical metrics from physical measurements [10,11,12,13,27]. Early models, which estimated visual acuity from various optical metrics, identified the MTFa from 0 to 50 lp/mm as the most effective predictor of visual acuity [12]. Since then, numerous authors have proposed alternative models using MTFa, extending these predictions to other psychophysical metrics, such as contrast sensitivity [11].

Prediction models are built establishing relationships between the mean psychophysical outcomes in a sample of eyes and the measurement of a single IOL. Therefore, the main purpose of the models is to predict the mean visual performance of a sample. Depending on the characteristics of the sample, the applicability of the model for different purposes will vary. This is because these models are empirical functions, not purely physical models, which makes them susceptible to biases from the clinical dataset they are based on. For example, the Vega et al. (2018) model exhibits a clear ceiling effect, likely due to the common practice in older studies of not measuring visual acuities above 0.0 logMAR. A key point of discussion when comparing these models is the reliance on either monocular or binocular defocus curve data, as seen in studies by Vega [10], Fernandez [11], and Armengol [13], which all used monocular data. In contrast, Alarcon’s model [12] utilized high-contrast binocular through-focus visual acuity. Due to the spatial summation of the visual system, binocular curves always result in visual acuities that are equal to or better than monocular ones in healthy eyes. This means that the prediction models based on monocular data show the minimum visual acuity that patients will achieve, as they will benefit from binocular summation in real life.

In our study, we evaluated a specific IOL with a power of 20 D, yielding outcomes closely aligned with those previously reported in the literature [18,28]. Can et al. observed for Acriva Trinova Pro C IOL, a decline in the MTF at a spatial frequency of 50 lp/mm for far distance as the pupil diameter increased [18]. Conversely, they noted an improvement in MTF at intermediate distances, with near-distance performance peaking at aperture sizes between 3.0 mm and 3.75 mm [18]. These trends were influenced by lens tilt and decentration, measured at 5° and 0.5 mm, respectively [18]. A limitation of their study was the reliance on a single spatial frequency (50 lp/mm) to assess optical performance.

In contrast, Labuz et al. reported MTFa and estimated visual acuity outcomes comparable to those in our study [28], although they limited their analysis to a 3.0 mm pupil using the Alarcon prediction model [12]. Their findings demonstrated smoother VADC than in our study, primarily attributable to differences in optical bench testing methods, including variations in SA and chromatic aberration induced by the cornea [17], +0.27 µm for SA and 1 D of chromatic aberration compared to the neutral correction in our study. Furthermore, their results indicated that the near peak of visual acuity was located at approximately 3 D of defocus; however, these defocus curves were not corrected to account for defocus lenses at the spectacle plane instead of the IOL plane [28]. The latter is a necessary step for functional classification of IOLs [8].

Although differences in optical characterization between Labuz et al. and our study did not alter the functional classification of the IOL at a pupil diameter of 3 mm (Full-DOFi Smooth) [28], these differences were significant enough to hypothetically affect outcomes for bigger pupil diameters. Specifically, visual acuity improved from intermediate to nearby 0.07 logMAR in their study and by 0.12 logMAR in ours. This shift in the magnitude of the difference between the intermediate and near vision influenced the functional classification of the IOL in our study, where the optical bench setup predicted a transition to a Full-DOFi Steep response for pupil diameters ≥3.5 mm using the Alarcon prediction model [12]. In contrast, the Labuz et al. optical bench setup would likely maintain the IOL classification as Smooth over a broader range of pupil diameters. Unfortunately, the predicted VADC in their study was only reported for 3 mm even though measurements were also taken for 4.5 mm. In addition, the original Alarcon model had two key limitations. First, it estimated binocular instead of monocular visual acuity [22], whereas functional classification should be based on the monocular visual acuity defocus curve [8,12]. Second, the model was fitted for a 3 mm pupil size, questioning the applicability of this model for computing the performance at particular pupil sizes [22]. Both limitations were recently addressed, evidencing that current prediction models can be effectively applied to predict visual performance for different pupil sizes without needing fundamental changes, although with less accuracy for smaller pupil sizes [29]. In addition, Vega et al. also described a progressive overestimation of visual acuity as we move away from the 0 D defocus point in the VADC for smaller pupil sizes, and underestimation for larger pupils when a single prediction model for a 3 mm of pupil is used for estimating the performance of an EDOF IOL [30]. Thus, the use of current prediction models to functionally classify pupil dependency should be approached with caution, as existing evidence has highlighted their limitations. Given that these models are still in the early stages of development and refinement, further advancements may be necessary to improve their predictive accuracy. Moreover, their clinical relevance remains uncertain, as few have been prospectively validated against actual patient data under standardized conditions, either with IOLs different from those for which the model was originally designed or under varying conditions, such as changes in pupil diameter addressed in this study. Establishing such validation is essential to confirm their robustness and to support their integration into routine clinical decision-making. Furthermore, future inclusion of prediction confidence intervals for the model fitting would strengthen the interpretation.

Beyond differences in functional classification attributed to the optical bench setup, our study demonstrated that this classification also depends on the prediction model employed (see Table 1). Prediction models developed using Partial and Full-DOFi IOLs spanned an MTFa range of up to 40, necessitating non-linear fitting (Figure 2) [10,12]. In contrast, models constructed exclusively with Full-DOFi IOLs permit linear fitting, as their MTFa values do not exceed 30 [11]. This broader range (up to 40) compared to the narrower range (30) explains the functional classification shift observed in our optical bench setup for all models [10,12,13], except for the Fernández et al. models [11].

Our study has limitations that have been previously described. One limitation is the use of a non-physiological model eye for advanced IOL characterization (ISO1 model eye) [31]. This simplified model, while allowing us to compare our results with other studies that adhere to this standard, does not fully replicate the physiological conditions of the human eye. The ISO1 corneal model has minimalSA, and these factors can differ from those employed in developing the visual acuity prediction models used in the study. While our study did not evaluate SA, we recognize that it is a crucial parameter for visual quality, especially with large pupils, and its analysis will be fundamental in future research. In a realistic cornea, a positive SA would cause defocus curves to widen and visual acuity to decrease at the focal points. This effect would be most noticeable with larger pupils (e.g., above 4.0–4.5 mm) where the influence of SA is more prominent. Additionally, while the IOL is clearly classified as a Full-DOFi Smooth for a standard pupil diameter of 3 mm, caution is needed when qualifying its functional classification as Full-DOFi Smooth or Steep for larger pupil diameters.

Despite these limitations, the functional classification of IOLs provides an evidence-based framework for defining commonly used concepts in the field without defined cut-offs up to date, such as categorizing an IOL as pupil- or non-pupil-dependent. From this perspective, a ‘pupil-dependent’ response refers to a pupil change that shifts an IOL from one functional category to another among the six possible classifications [7,8]. However, since this functional pupil-dependency is influenced by the optical bench setup and prediction model, and because bias can be produced by using a model fitted considering a single pupil size [29,30], some caution should be exercised in interpreting pupil-dependency using current prediction models. Misclassification of IOL performance across pupil diameters may lead to misinterpretation of key patient-reported outcomes, such as spectacle dependence. This is particularly relevant as each classification carries a different probability and certainty of achieving spectacle independence, which often represents the primary reason patients choose a SVLs [32].

## 5. Conclusions

When adequate clinical data is lacking for making decisions regarding the performance of an IOL based on biometric factors, such as pupil diameter, optical bench measurements provide a valuable tool for understanding IOL behavior. However, a disconnect remains between these measurements, typically expressed in optical metrics, and the clinical interpretation of their impact on functional visual performance. Prediction models bridge this gap by translating optical bench data into clinically meaningful metrics for both surgeons and patients. This approach enables the following: evaluation of optical properties under varying pupil diameters in a laboratory setting, estimation of the corresponding clinical defocus curve, and functional classification of each defocus profile. Consequently, it becomes possible to provide evidence-based support for commonly used clinical and marketing terms, such as “pupil-dependent,” which until now lacked objective criteria. Using this functional classification, an IOL may be qualified as pupil-dependent if its functional response changes classification with variations in pupil size. Nevertheless, as demonstrated in our study, current prediction models have inherent limitations that can lead to discrepancies between estimated and actual clinical outcomes. Moreover, the specific optical bench setup used to derive the MTFa values can significantly influence the reliability of these outcomes and prediction models fitted with a mean single pupil size can result in bias in smaller and larger pupils. Future models may improve accuracy by focusing on specific functional IOL categories rather than attempting to generalize across diverse defocus profiles, correcting potential sources of bias, and incorporating multiple pupil diameters into the prediction. Until such refinements are realized, caution should be exercised when interpreting outcomes derived from models based on methods that differ from those used during their development.

## Figures and Tables

**Figure 1 diagnostics-15-02446-f001:**
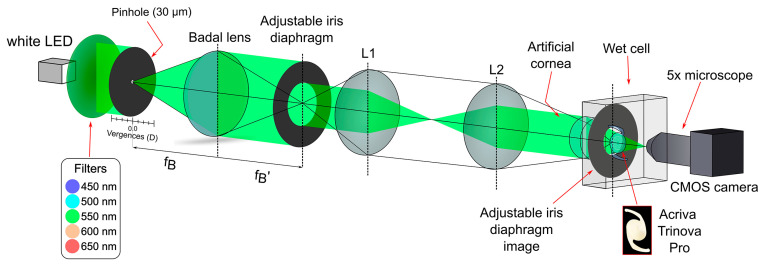
Optical bench setup representation.

**Figure 2 diagnostics-15-02446-f002:**
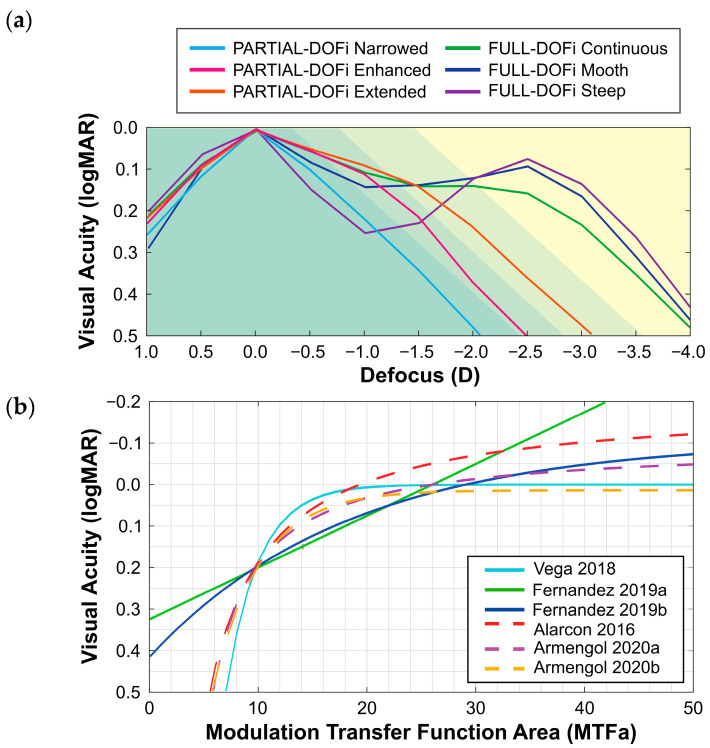
(**a**) Mean defocus curve obtained for each IOL classification. The background color indicates the different depth-of-field (DOFi) ranges according to functional classification: Partial (green, with progressively decreasing tone for Narrowed, Enhanced, and Extended) and Full (yellow); (**b**) Relationship between visual acuity and modulation transfer area for each one of the prediction models used in the study.

**Figure 3 diagnostics-15-02446-f003:**
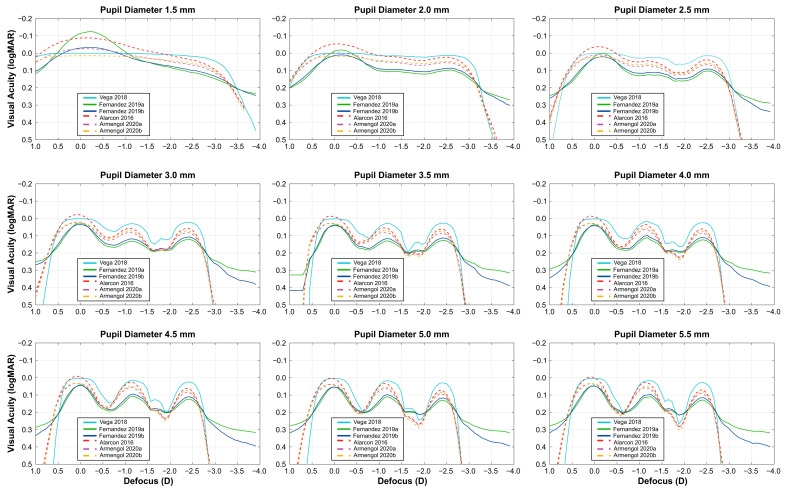
Predicted visual acuity defocus curves for different pupil diameters (described at the top) and prediction model (described inside the legend).

**Table 1 diagnostics-15-02446-t001:** Difference in visual acuity from the minimum inflexion point at intermediate vision (defocus range −0.50 to −2.00 D) and maximum at near (defocus range −2.00 to −4.00 D).

	Monochromatic Light Prediction Models			Polychromatic Light Prediction Models		
Pupil (mm)	Vega 2018	Fernández 2019a	Fernández 2019b	Alarcón 2016	Armengol 2020a	Armengol 2020b
1.5	0 ^a^	0 ^a^	0 ^a^	0 ^a^	0 ^a^	0 ^a^
2.0	0 ^a^	0 ^a^	0 ^a^	0 ^a^	0 ^a^	0 ^a^
2.5	0.04 ^a^	0.06 ^b^	0.06 ^b^	0.07 ^b^	0.06 ^b^	0.06 ^b^
3.0	0.13 ^b^	0.07 ^b^	0.07 ^b^	0.12 ^b^	0.11 ^b^	0.12 ^b^
3.5	0.17 ^c^	0.07 ^b^	0.09 ^b^	0.15 ^c^	0.13 ^b^	0.14 ^c^
4.0	0.14 ^c^	0.08 ^b^	0.08 ^b^	0.16 ^c^	0.15 ^c^	0.17 ^c^
4.5	0.17 ^c^	0.08 ^b^	0.09 ^b^	0.18 ^c^	0.16 ^c^	0.18 ^c^
5.0	0.21 ^c^	0.08 ^b^	0.09 ^b^	0.20 ^c^	0.18 ^c^	0.21 ^c^
5.5	0.24 ^c^	0.09 ^b^	0.10 ^b^	0.21 ^c^	0.20 ^c^	0.22 ^c^

^a^ Full-DOFi Continuous (blue cells). Difference between intermediate and near <0.05 logMAR. ^b^ Full-DOFi Smooth (green cells). Difference between intermediate and near ≥0.05 to <0.14 logMAR. ^c^ Full-DOFi Steep (orange cells). Difference between intermediate and near ≥0.14 logMAR.

## Data Availability

The data presented in this study are available on request from the corresponding author.

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
