# Peer review of "Reliability of Prediction Models for the Functional Classification of a Sinusoidal Intraocular Lens Depending on Pupil Diameter"

_diagnostics, 2025, doi:10.3390/diagnostics15192446_

Round 1
Reviewer 1 Report
Comments and Suggestions for Authors
See attached file.

Author Response
We extend our sincere gratitude to the reviewers for their constructive and insightful comments. We have carefully addressed each point, and the following document outlines our detailed responses and indicates the corresponding modifications made to the manuscript, with specific line numbers provided for clarity.
Reviewers' comments are presented in italics, while our responses are in normal font.
Reviewer 1
Areas for Improvement
- Clarity on Benchmarking and Model Validation
1.1 The paper effectively contrasts different prediction models, but the discussion would benefit from deeper reflection on the clinical validation status of each model. For example, have any of these models been prospectively tested against actual patient data using the same IOL (Acriva Trinova Pro C)?
Thank you for your feedback. We completely agree with this statement and have added a comprehensive discussion on the clinical validation status of each model in the manuscript (lines 240 to 251).
1.2 In Table 1 and associated figures, the use of statistical tools (e.g., confidence intervals, inter-model variability quantification) would strengthen the comparison.
We agree with your constructive comment. Unfortunately, since prediction models do not provide confidence intervals, only the mean sample visual acuity can be predicted. We have included this point in the discussion for future developments (lines 293-304).
- Visual Representation of Classification Transitions
2.1 Figure 2 could be significantly improved in terms of resolution and graphical editing.
Thank you for the feedback. We have revised Figure 2 and Figure 3 to improve their resolution and graphical quality. Additionally, we have updated the caption for Figure 2 to include a more detailed explanation (lines 179-181).
2.2 Consider adding a summary figure or heatmap showing the classification of each model across pupil diameters (e.g., a color-coded matrix).
Thank you for your valuable comment. We agree that a clear visual representation, such as a heatmap, is an excellent way to show the differences in the IOL's functional classification.
In our manuscript, Table 1 was designed to serve as a visual summary of the results, classifying the functional response of each model based on pupil diameter. To further enhance the visualization and make the table more intuitive, we have implemented your suggestion. The table now uses a more distinct color differentiation: blue for the Continuous classification, green for Smooth, and orange for Steep. Additionally, we have updated the table footnote (lines 217-219) to not only reference the superscripts (a, b, and c) but also to explain the meaning of each color. We believe these changes significantly improve the clarity and effectiveness of the table, making it easier for the reader to interpret the transitions in functional classification.
- Bench-to-Clinic Translation
3.1 While the paper rightly highlights model biases based on optical setup differences, it would benefit from a deeper discussion of potential clinical consequences. For example, how might a misclassification (e.g., Smooth vs. Steep) influence IOL selection for patients with large scotopic pupils?
This has been clarified in the manuscript (lines 336-340).
3.2 The discussion could briefly touch on the relevance of monocular vs. binocular defocus curves and how these may impact model reliability in bilateral IOL implantation contexts.
Thank you for your feedback. We have incorporated a deeper discussion on the relevance of monocular versus binocular defocus curves into the manuscript (lines 240-251).
The models are based on clinical defocus curve measurements, but it is crucial to differentiate between monocular and binocular curves, as this may impact the reliability of the model in the context of bilateral IOL implantation. Most of the prediction models are fundamentally based on monocular data: Vega (2018), Armengol (2020) (models a and b), and Fernández (2019). On the other hand, Alarcon's model differs by using high-contrast binocular through-focus visual acuity.
It is important to note that due to the spatial summation of the visual system, binocular defocus curves always result in visual acuities that are equal to or better than monocular ones in healthy eyes. Therefore, the prediction models based on monocular data (with the exception of Alarcon's) show the minimum visual acuity that patients with this IOL will achieve, since in real life they will benefit from binocular summation. We hope this addition clarifies the point and strengthens our discussion.
- Terminological Consistency
4.1 Use of “DOFi” vs. “RoF” could be standardized or clarified further, as both terms are used across prediction model definitions and clinical metrics. A simple figure summarizing the taxonomy (Partial vs. Full RoF; Continuous/Smooth/Steep) would be pedagogically valuable for readers unfamiliar with this framework.
At the ESCRS Congress 2025, held this week in Copenhagen, the ESCRS, ASCRS, APACRS, and LATAMSCRS societies agreed to adopt the uniform use of the term DOFi. Accordingly, DOFi has been used consistently throughout this manuscript. The legend of Figure 1 pedagogically illustrates the different DOFi ranges that can be achieved.
- Model Assumptions and Limitations
5.1 The paper acknowledges the limitations of using prediction models trained on a single pupil diameter, but additional sensitivity analysis showing model behavior with alternative optical parameters (e.g., spherical aberration or decentration) would enhance robustness.
We sincerely thank the reviewer for their valuable comment. We agree that additional optical parameters, such as spherical aberration or lens decentration, are crucial for a complete understanding of visual performance.
For this study, we chose to focus specifically on the impact of pupil diameter, as it is a key and variable parameter for the performance of multifocal IOLs. To this end, we conducted a comprehensive evaluation by measuring performance across nine different pupil diameters, which we consider a robust sensitivity analysis in itself.
As suggested by the reviewer, we have added a sentence in the manuscript acknowledging the critical importance of spherical aberration and our intention to evaluate it in future research to further enhance the applicability of our prediction models (lines 319-321). Adding an evaluation of other optical parameters would have significantly increased the study's complexity, both in data acquisition and model construction. However, we recognize the importance of this analysis and plan to include an evaluation of these parameters in future studies.
5.2 The ISO1 model eye is a known limitation—consider briefly comparing results against a physiological model eye as discussed in [32].
We appreciate the reviewer's valuable comment. We agree that the ISO 1 model eye used in our study is a known limitation, as it is a simplified eye model with corrected spherical aberration. Both the ISO 1 and ISO 2 models do not incorporate human chromatic aberration. The ISO1 model was used to ensure standardization and comparability with previous studies in the literature. As the reviewer rightly points out, the use of more advanced physiological eye models would have provided results closer to actual clinical conditions. These models present a positive spherical aberration, which could cause defocus curves to widen and visual acuity to decrease at the focal points, an effect that would be more noticeable with large pupils.
We have added a paragraph to the Discussion section (lines 315-324) to address this limitation and explain how the use of a more realistic physiological eye model could influence the measurements.
Reviewer 2 Report
Comments and Suggestions for Authors
The article addresses the topic of using optical bench in-vitro measurements to predict postoperative visual acuity. I consider this subject very important in order to communicate optical data to ophthalmologists and thus assist them in selecting the lens that best suits the patient’s needs and characteristics. Despite the models proposed in the literature, I agree that the results must be interpreted with extreme caution. The authors present an experimental study in which they set up a custom experimental setup, with reasonable limitation compared other state-of-arts instruments, that essentially measures the PSF by simulating a point source with a pinhole (with diameter small enough for the required resolution in term of spatial frequencies), and subsequently calculating the MTF and the MTFa.
Some questions:
1) In Alacorn (2016), what I found particularly interesting is the weighted MTF, i.e., with a limit of 150 lp/mm and the use of the threshold contrast sensitivity. Why did the authors not consider this metric?
2) In Armengol (2020), the metric based on MTFa is compared with one based on energy efficiency (EE). Since the latter can be retrieved from the PSF, this metric could also be obtained from the measurements in the setup designed by the authors. Indeed, the PSF, not the MTF, is directly measured. Did the authors take this into account? Is the absence of this metric due to the desire for homogeneity with the others, or was it neglected because the pinhole size might be too large?
3) The authors state that the specific optical bench setup can influence reliability and thus prediction. However, I believe that an experimental measurement in optics should be defined by its accuracy and precision if the same quantity is measured. In my view, Although these models have considerable importance, the models proposed so far are dependent on the clinical data of the sample, since they are not physical models but rather empirical functions fitted to a sample. For example, the Vega (2018) model appears unable to predict visual acuity above 0 logMAR, regardless of the MTFa values. This seems to be a bias caused by the clinical dataset used, which may act as a cutoff. Do the authors have any opinion on this?
4) In my opinion, some figures need an improvement and they are not acceptable in the present form:
- the resolution seems often low (probably because a resizing)
- some figures are boxed, others not (for example the Fig. 2A).
- The font sizes are inconsistent across figures, and in some instances appear excessively small
Author Response
We extend our sincere gratitude to the reviewers for their constructive and insightful comments. We have carefully addressed each point, and the following document outlines our detailed responses and indicates the corresponding modifications made to the manuscript, with specific line numbers provided for clarity.
Reviewers' comments are presented in italics, while our responses are in normal font.
Reviewer 2
Comments and Suggestions for Authors
The article addresses the topic of using optical bench in-vitro measurements to predict postoperative visual acuity. I consider this subject very important in order to communicate optical data to ophthalmologists and thus assist them in selecting the lens that best suits the patient’s needs and characteristics. Despite the models proposed in the literature, I agree that the results must be interpreted with extreme caution. The authors present an experimental study in which they set up a custom experimental setup, with reasonable limitation compared other state-of-arts instruments, that essentially measures the PSF by simulating a point source with a pinhole (with diameter small enough for the required resolution in term of spatial frequencies), and subsequently calculating the MTF and the MTFa.
Some questions:
1) In Alacorn (2016), what I found particularly interesting is the weighted MTF, i.e., with a limit of 150 lp/mm and the use of the threshold contrast sensitivity. Why did the authors not consider this metric?
We thank the reviewer for their insightful comment and for pointing out the weighted MTF metric used by Alarcón (2016). The decision not to use this metric is based precisely on Alarcón's own conclusions. In that study, the author found that no significant differences were found between the weighted MTF (wMTF) and the standard MTF-based metrics (MTFa). Given that both metrics were shown to be equivalent in that context, we chose to use MTFa as it is a simpler metric to calculate and is the most commonly used in current ophthalmological literature for characterizing intraocular lenses (in fact, this metric was also included recently in the ISO 11979-2:2024 Standard). Therefore, this choice allowed us to maintain consistency with widely accepted methods in the literature and to simplify the characterization without losing crucial information, as validated by Alarcón.
2) In Armengol (2020), the metric based on MTFa is compared with one based on energy efficiency (EE). Since the latter can be retrieved from the PSF, this metric could also be obtained from the measurements in the setup designed by the authors. Indeed, the PSF, not the MTF, is directly measured. Did the authors take this into account? Is the absence of this metric due to the desire for homogeneity with the others, or was it neglected because the pinhole size might be too large?
We appreciate the reviewer's detailed comment. We agree, the energy efficiency (EE) metric could have been calculated from the point spread function (PSF) we obtained from our measurements, But, in fact, as we mentioned our decision not to include the EE metric was due to the need to homogenize the results for comparing the applicability of all prediction models in our study. Therefore, to be able to apply the same metrics to all prediction models and simplify the comparison, we chose to use only MTFa, as it is a metric common to all of them.
On the other hand, yes, we think that a 200 mn pi-hole as the one employed in Armengol (2020) might be too large.
3) The authors state that the specific optical bench setup can influence reliability and thus prediction. However, I believe that an experimental measurement in optics should be defined by its accuracy and precision if the same quantity is measured. In my view, Although these models have considerable importance, the models proposed so far are dependent on the clinical data of the sample, since they are not physical models but rather empirical functions fitted to a sample. For example, the Vega (2018) model appears unable to predict visual acuity above 0 logMAR, regardless of the MTFa values. This seems to be a bias caused by the clinical dataset used, which may act as a cutoff. Do the authors have any opinion on this?
Thank you for your insightful and accurate comment. We agree that the models proposed to date are empirical functions that depend on the clinical data of the sample they are based on, rather than being purely physical models. As suggested, we have added a sentence in the discussion section (lines 240-243) acknowledging that these models can present biases.
While the main goal of our study was not to critique these models, we also noted the key issue you identified: the Vega (2018) model exhibits a clear ceiling effect, which is visible in our Figure 2B. This effect is also present, to a lesser extent, in other models, such as Armengol B. We believe this bias could be due to the common clinical practice, especially in older studies, of not measuring visual acuity beyond 0 logMAR (or 1.0 decimal VA), which acts as an artificial cutoff in the dataset.
We firmly believe that while these models are based on empirical data, they can be improved to avoid limitations such as this ceiling effect. For example, this bias is no longer present in some more recent models.
4) In my opinion, some figures need an improvement and they are not acceptable in the present form:
- the resolution seems often low (probably because a resizing)
- some figures are boxed, others not (for example the Fig. 2A).
- The font sizes are inconsistent across figures, and in some instances appear excessively small
Thank you for your feedback. We have revised the figures to address the concerns regarding their quality. Both figures have been improved in terms of resolution and consistency. The font sizes are now consistent across all figures. The slight difference in perceived font size between Figure 2 and Figure 3 is due to the fact that Figure 3 contains nine subfigures, which required a reduction in the overall scale of the elements within the figure to maintain the aspect ratio and ensure all information is clearly displayed. We have also ensured that both figures are formatted consistently in the manuscript, including the use of boxing.
Round 2
Reviewer 2 Report
Comments and Suggestions for Authors
In my opinion the authors clarified the raised points, in particular concerning the problem of the used dataset in the previous models used in this work. I agree with the authors that a common issue in some dataset is the practice to consider 0 logMar as the limit maximum value for the visual acuity.